# The Role of Near-Field Communication Mobile Payments in Sustainable Restaurant Operations: A Restaurateur's Perspective

**Wen-Way Yu and Chin-Yi Fang ***

Graduate Institute of Sport, Leisure and Hospitality Management, National Taiwan Normal University, Heping E. Rd., Taipei City 106, Taiwan; tim-yu@wowmiss.com
* Correspondence: fred@ntnu.edu.tw or chinyifang@gmail.com; Tel.: +886-2-7749-1448

**Abstract:** Mobile payments have emerged as a viable alternative to cash and credit cards and are rapidly gaining popularity worldwide. Limited research has explored the effects of mobile payments on restaurant performance from the perspective of restaurateurs. This study utilized a combination of the innovation diffusion theory (IDT) and the technology acceptance model (TAM) to investigate the influence of a near-field-communication (NFC) mobile payment environment on restaurant operating performance (ROP). Through convenience sampling, questionnaires were distributed to restaurant owners and managers, resulting in 279 valid responses. The empirical findings revealed that sales growth ($\beta$ = 0.478), cost savings ($\beta$ = −0.236), flexibility ($\beta$ = 0.117), accessibility ($\beta$ = 0.184), and trust and safety ($\beta$ = 0.286) significantly impacted ROP. When considering restaurant size as a moderator for analysis, only two constructs, namely, accessibility ($\beta$ = 0.108) and trust and safety ($\beta$ = −0.160), showed significant impacts on ROP. These empirical insights offer valuable references to restaurateurs for enhancing ROP by leveraging the mobile payment environment.

**Keywords:** restaurant operating performance; restaurant size; innovation diffusion theory (IDT); technology acceptance model (TAM)

## 1. Introduction

### 1.1. Research Background

With the rise of mobile devices and their integration into various aspects of our lives, including personal banking and powerful apps, people are increasingly relying on them to manage their daily activities. Bittman et al. [1] noted that mobile phones have rapidly become one of the most widely adopted devices in recent technological revolutions. As a result, near-field communication (NFC) mobile payment environments, which facilitate transactions via smartphones, are gaining popularity. However, from a business owner's perspective, do NFC mobile payments offer a competitive advantage or create operational challenges in fast-paced environments? Empirical research on the impact of NFC environments on small business performance, particularly in the food and beverage industry, is limited. Nevertheless, more and more customers are accustomed to making payments with their mobile devices. The majority of current research focuses on examining the influence of mobile payment on customer satisfaction [2], intention [3], or loyalty [4]. Limited research has explored the effects of NFC mobile payments on restaurant performance from the perspective of restaurateurs. Thus, this paper's key contribution stems from its integration of the overarching innovation diffusion theory (IDT) model with the technology acceptance model (TAM). This integration aims to analyze how the NFC mobile payment environment influences the operational performance of restaurants (ROP) from the restaurateurs' viewpoint.

The World Health Organization (WHO) suggested that global consumers adopt contactless payment modes. This further helped raise the mobile payment transaction rate

and boosted contactless payment modes [5]. Nowadays, two major technological layers, NFC and barcode, are on the market as contactless communication technologies for mobile payment. NFC-based payments outperform barcode-based payments in terms of security, usability, and convenience [6]. The NFC mobile payment is estimated to be $4.8 billion in 2015 and is expected to reach 47.42 billion by 2024 [7]. This demonstrates the increasing importance of NFC in the era of smartphones. Advanced environments are emerging, facilitating the installation of NFC mobile payment. Mobile phones have become a convenient and preferred platform for various activities, surpassing traditional desktop computers. The maturation of cloud computing, smartphone technology, and communication advancements has propelled the rapid adoption of mobile payments worldwide, replacing cash and credit cards [8]. The COVID-19 pandemic further accelerated the adoption of mobile payments as consumers sought safer payment methods, leading to a surge in usage even after lockdown measures were lifted [9]. Reports suggest that non-cash transactions, including mobile application-based transactions, are expected to increase from 17% in 2021 to 28% in 2026, driven by endorsements from the business and retail sectors. These sectors recognize the advantages of mobile payments, such as cost reduction and increased consumer spending on products and services [10].

However, The Wall Street Journal [11] reported that mobile payment operators have observed a lack of compelling reasons to encourage small-business retailers to adopt mobile payments so far. Among the factors deterring merchants from embracing mobile payments, there are technological incompatibility, complexity, the investment costs involved, and the absence of sufficient critical mass and knowledge in this domain [12]. Nonetheless, Begonha et al. [13] suggested that NFC mobile payment environments can offer a convenient and cost-effective alternative for merchants who do not typically accept credit card payments. Researchers have also proposed that the food and beverage industry can enhance customer loyalty and satisfaction by implementing technology-driven models that improve quality, efficiency, and the service–profit chain for performance growth [13,14].

### 1.2. Research Objectives

Given that the majority of restaurants in Taiwan are unlisted companies and obtaining secondary data on their financial performance is challenging, this study aims to employ a survey method to achieve the following objectives: (1) examine the moderating effects of restaurant size on the relationship between the mobile payment environment and restaurant operational performance (ROP); and (2) identify the influential factors of the NFC mobile payment environment on ROP by combining the Innovation Diffusion Theory (IDT) and Technology Acceptance Model (TAM).

### 1.3. Mobile Payment Environment Definition and Literature Review

Au and Kauffman [15] defined a mobile payment environment as a mobile device initiating authorization for payments and confirming the exchange of value for goods and services. Mallat [16] defined it as conducting funds transfer or payment transactions with a mobile device, either through a third party or direct payment to the receiving party. Ghezzi et al. [17] recapped the concept of a mobile payment environment as an electronic payment procedure in which at least one part of the transaction is conducted using a mobile phone capable of securely handling a financial transaction over a mobile network, or via various wireless technologies, such as NFC or Bluetooth. Mobile payment environments can be divided into remote payments and short-range NFC payment contactless technologies [14]. The technology and platforms used by these two forms of payment are different. A remote payment environment is an e-commerce online transaction, in which consumers use their mobile phone to make payments and complete the shopping procedures on the Internet through credit cards, IC cards, or electronic coupons. The NFC payment environment uses a mobile phone as the payment tool, which is used in a physical store to complete the payment transaction in a connected or offline mode [18]. NFC lets two devices positioned at a very short distance from each other exchange data. Both devices ought to

be equipped with an NFC chip [19]. Some scholars have already recognized NFC as the future trend in mobile payment environments [20]. This is because the advantages of the NFC environment are low power, accessibility, and simple communication equipment [21]; also, NFC technology does not require complex device pairing. Therefore, it provides many benefits to food and beverage operators and consumers [22,23]. Hayashi [24] pointed out that the environment using NFC mobile payment is 15 to 30 s faster than usual card swiping. This is because the time spent on the NFC device is in short, simple operating method and secures message transfer; this type of payment technology is most convenient and suitable for fast-paced restaurants and travel environments [21,25]. Getz and Robinson [26] also mentioned that it is possible to use NFC mobile payment environment to raise consumer satisfaction towards the restaurant. Slade et al. [27] argued that mobile-payment-related research is still in its early stage, even though there has been a relative increase in mobile payment research over the past few years [28]. However, as compared to the extensive e-commerce research (e.g., online bank, mobile bank, etc.), the mobile payment environment is a relatively new area of research [29], with most of the research focused on the consumer perspectives [30–33]. There are quite rare studies to understand the environment that merchants use for mobile payments and the method they acquire to implement this new payment vehicle [34]. Not to mention that little research examines the influential factors in the mobile payment environment from the restaurateurs' perspective [35].

### 1.4. The Impact of Mobile Payment Environment on Operations Performance

Niedritis et al. [36] indicated that effective business processes ensure the achievement of the enterprise's goals. The performance measurement should be performed from different perspectives. Performance is the extent to which the organization's goal is achieved [37]. To meet an organization's goal and to create the organization's value, an organization needs to establish a performance measurement scheme and create a revenue-generating environment. Fredendall and Robbins [38] believe that the purpose of an organization's existence is to achieve its predetermined goal, performance is to measure the extent to which the organization's goal is achieved, and managing performance is thus the achievement rate for a business's strategic goal. Organizations effectively make an empowering environment, or latently adapt to the environment through asset distribution; the methods taken to accomplish the association's objective are an essential record to survey if a business activity has been fruitful [39], i.e., the mission of the manager of a business is to develop an environment that increases organizational performance and creates maximum efficiency with the least investment. Qiu [40] believes that the subject of performance evaluation is not an individual of the organization; it should be the overall organizational performance. Performance evaluation is the systematic process of how an organization achieves its goals.

There are different perspectives on performance measurement indicators among different scholars. The objective measurement using secondary data for listed companies is one commonly used method. Miller and Friesen [41] proposed various performance evaluation indicators, including investment returns, cash flow, market share stability, price-to-book ratios, and employee productivity. Woo and Willard [42] suggested that there are 14 types of performance measurement indexes, including investment returns, sales returns, sales income, cash flow, investment, etc. Walker and Ruekert [43] used three indexes, financial performance, growth, and profitability, as benchmarks to measure the overall operations performance of a company. Richard and Johnson [44] suggested that objective measurement of business performance could use employee turnover rate, employee productivity, and return on equity as a basis.

The second perspective for performance measurement is using subjective assessment, commonly for unlisted small-and-medium enterprises. Gunday et al. [45] brought up the survey method in performance measurement. Moideenkutty et al. [46] highlighted that the use of questionnaires could reflect respondents' feelings, which are the subjective performance measurement. Amin et al. [47] also used the questionnaire method as the subjective measurement of business performance. The subjective measurement method

as compared to the objective measurement method enables a higher probability to receive more information about the organization they served [46,47]. Due to the fact that this research sample focused on unlisted restaurants, the ROP used the questionnaire for measuring business performance, as proposed by Gunday et al. [45].

*1.5. Theoretical Model and Research Hypotheses*

Pal et al. [48] used a keyword searching method to review a total of 50 pieces of literature about the mobile payment environment and mobile banks in recent years; most of the papers attempted to utilize a theoretical model to investigate the determinants of consumers' intentions of mobile payment. Kim et al. [49] claimed that it would be better to use TAM than the unified theory of acceptance and use of technology (UTAUT) proposed by Venkatesh et al. [50] as a theoretical basis for research about mobile payment environments. Andersson [51] also pointed out that most research used these two theories, TAM and IDT, to explore the drivers of consumers' uses for new information technology environments.

Davis [52] developed the TAM theory to explain the decision-making factors in accepting an information technology environment, with a particular focus on technology use behavior. The implication is that the individual level of willingness to accept the new technology environment is dependent on the individual perceived usefulness and perceived ease of use for this technology. Szajna [53] and Wu and Wang [54] suggested that TAM needed to integrate with other variables in order to increase the model's explanatory power.

IDT proposed by Rogers [55] is used to explain diffusion behavior. New innovation needs to undergo specific communications channels and is accepted by users as time goes on; this is so-called diffusion behavior. Five cognitive constructs of innovation are (1) relative advantage; (2) compatibility; (3) complexity or accessibility; (4) trialability (the degree the users may be tried to use before adoption); and (5) observability. Tornatzky and Klein [56] researched 57 papers about innovative diffusion and found that only three innovative characteristics had a significant influence on consumers' decision-making for adopting innovation. Therefore, some research related to innovation adoption only focused on these three variables that influenced adoption behaviors [57,58]. The three variables were comparative advantage, compatibility, and accessibility.

Moore and Benbasat [58] conducted semi-structured interviews with managers whose environments had adopted information technology innovation and obtained 143 valid questionnaires. They used exploratory factor analysis and extracted six factors for mobile payment environment adoption. They were sales growth, cost reduction, flexibility, accessibility, trust and safety, and network externalities. The literature also pointed out that the relative advantage construct included the sub-constructs of sales growth and cost saving. Sheikh et al. [59] used the questionnaire method from 278 marketing managers to validate the positive impact of relative advantage on the performance of the textile business in Pakistan. The NFC–mobile payment environment would benefit from eliminating to use of cash, offering fast speed and convenience, and exchange of secure data between devices in environments with a high volume of payments, such as restaurants [21,29]. Both merchants and consumers benefit from operation time reduction, with feasible cost savings and productivity gains [29]. According to the survey conducted by Statista [60], the worldwide mobile payment revenue in 2015 was USD450 billion and is expected to exceed USD1 trillion in 2019, thus becoming one of the most important environments for conducting mobile transactions.

Hence, this paper establishes the following hypothesis:

**H1:** *Relative advantage environment–sales growth has a positive influence on ROP.*

**H2:** *Relative advantage environment–cost saving has a positive influence on ROP.*

As compared to traditional e-commerce, the most important quality of the mobile payment environment is flexibility. It is the ability to use mobile network functions at any time and any place, providing more services and functions for the users [61,62]. Flexibility is also recognized as one of the most important factors for the success of the mobile commerce

environment [63]. Moore and Benbasat [58] pointed out that the attractive factor for mobile payment users also included flexibility. H3 was established as below:

**H3:** *Flexibility has a positive influence on ROP.*

Research has proven that accessibility is a very important aspect in influencing consumers to use new technology [64,65]. As the mobile payment environment can provide a greater scope of payment capability, consumers intend to use mobile payment [49]. Dahlberg et al. [66] also claimed that accessibility is the most important factor in the mobile payment environment.

**H4:** *Accessibility has a positive influence on ROP.*

In spite of the fact that innovation advancements realized numerous advantages for buyers, however, there are still a few factors that could hinder customers' acknowledgment of the technology innovation. Past literature indicated that new technology often comes with certain risks [67]. Security is one of the key factors in the acceptance of a new technological environment [68]. Whether consumers are willing to use the Internet to conduct transactions, primary consideration is given to transaction security [69], i.e., the more secure the online transaction environment the more the consumer is willing to use online transactions. Chang et al. [70] suggested that consumers' and merchants' payment services rely heavily upon a secure and reliable payment environment, even if it is easy to use. According to research by Bast [71], restaurants that use NFC mobile payment environments also heavily rely on system security. This paper establishes the following hypothesis:

**H5:** *An environment with trust and safety has a positive influence on ROP.*

Melitz and Ottaviano [72] and Rumelt [73] revealed that company size is an important moderator for a company's operational performance, and, as compared to small and medium businesses, large companies have advantages in terms of market, management, and financial resources [74]. Therefore, according to past literature [75,76], this research hypothesizes that the size of the company would have different influences on the relationship between five constructs and ROP as below,

**H6a:** *Company size moderates the relationship between sales growth and ROP.*

**H6b:** *Company size moderates the relationship between cost saving and ROP.*

**H6c:** *Company size moderates the relationship between flexibility and ROP.*

**H6d:** *Company size moderates the relationship between accessibility and ROP.*

**H6e:** *Company size moderates the relationship between trust and safety and ROP.*

## 2. Method

### 2.1. Research Framework and Methodology

This paper combined TAM and IDT and added two important variables (flexibility and trust and safety) as a research framework in Figure 1 to examine the determinants of NFC mobile payment to ROP. We adopted two constructs (perceived usefulness and perceived ease of use) from TAM, whereas the other constructs (relative advantage and accessibility) were from IDT. Meanwhile, we combined questionnaire items from the construct "perceived ease of use" from TAM and "accessibility" from IDT into one construct. This paper referred to two works from Mallat and Tuunainen [77] and Moore and Benbasat [58] and mainly focused on examining the drivers and barriers for merchants using NFC mobile payment environments, not including compatibility, trialability, and observability from IDT. There are six constructs in Figure 1: sales growth, cost saving, flexibility, accessibility, and trust and safety as the independent variables, and ROP as the dependent variable. All constructs use the Likert-type 5-point scale as a measurement tool, with 1 for strongly disagree to 5 representing strongly agree.

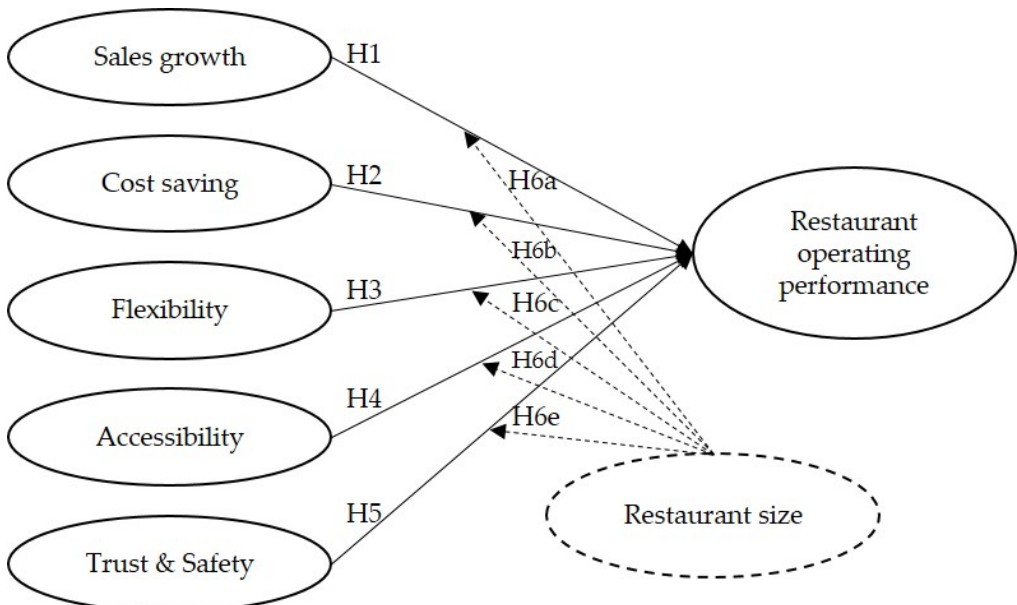

**Figure 1.** The research framework.

Once the survey dimensions were established and the translation of survey items was completed, a panel of eight experts, consisting of four academic professors and four senior-management-level professionals with over ten years of industry experience, was convened to assess the questionnaire's validity. The experts provided feedback on the accuracy, appropriateness, relevance, coverage, and wording of the questions for each dimension. After gathering and incorporating the majority of their evaluations, the suggestions given by the experts were used to semantically revise the study questions, ensuring that the survey content remained relevant and user-friendly. The pretest was distributed to 52 restaurant owners as the basis of the reliability test. After the items' modification from the experts' opinions and pilot test responses, a total of 279 valid questionnaires from restaurant owners and managers were collected. This survey used the Likert-type 5-point scale as a measurement tool, with 1 for strongly disagree to 5 representing strongly agree. The questionnaire was distributed to in-store managers or owners of catering businesses that already had mobile payment systems. Prior to sending the survey, the job positions of the respondents, whether managerial or cashier, were verified. Once confirmed, an electronic survey was sent to the participants, with a request to pass the survey along to other restaurateurs who had implemented mobile payment systems. Managerial staff were selected as participants because of their in-depth understanding of business operations, enabling them to provide accurate evaluations [78]. Additionally, cashiers, being key frontline personnel operating the mobile payment system, could offer objective responses to the survey questions.

### 2.2. Reliability Test of Pretest Questionnaire

The pilot survey was released via the Internet and distributing 54 copies of the pretest questionnaire. After deducting two copies of invalid questionnaires, a total of 52 valid questionnaires were obtained. SPSS 23.0 was used to conduct a reliability test for the questionnaire. Cronbach's α value was used to measure and test the internal consistency of each independent and dependent variable. Moideenkutty et al. [46] pointed out that it is acceptable for all dimensions with Cronbach's α value larger than 0.6. The Cronbach's α for all of the dimensions of this research was between 0.675 and 0.911. This shows that there was consistency and reliability for each dimension of this survey in Table 1.

**Table 1.** Reliability assessment of measurement model.

| Construct | Cronbach's $\alpha$ | No of Item |
|---|---|---|
| Sales growth | 0.911 | 7 |
| Cost saving | 0.860 | 7 |
| Flexibility | 0.675 | 3 |
| Accessibility | 0.927 | 5 |
| Trust and safety | 0.908 | 4 |
| ROP | 0.869 | 4 |

Note: ROP means restaurant operational performance.

## 3. Results

### 3.1. Descriptive Statistics

The effective sample size should be at least five times the minimal number of survey items [79]. There are six constructs, including 30 items, in this paper. Therefore, the adequate sample size should be at least 150 respondents. The required sample size depends on various factors in a study, such as the level of variability between variables and the degree of over-determination (i.e., the ratio of variables to the number of factors) of the factors [80]. A general guideline suggests having at least 10 participants for each scale item, resulting in an ideal respondent-to-item ratio of 10:1 [81]. However, some researchers have proposed sample sizes that are not directly tied to the number of survey items. Clark and Watson [82] suggested using 300 respondents after initial pre-testing. Additionally, others have recommended a range of 200–300 participants as suitable for factor analysis [83,84]. Thus, distributing this survey to 300 respondents was appropriate [85,86]. After deducting incomplete invalid samples, a total of 279 valid questionnaires were obtained. SPSS 23.0 was used to analyze descriptive statistics, as shown in Table 2.

**Table 2.** Descriptive statistics.

| Item | Class | Number (People) | Percent (%) |
|---|---|---|---|
| Gender | Male | 142 | 50.9 |
|  | Female | 137 | 49.1 |
| Age | <25 years | 7 | 2.5 |
|  | 26–30 years | 50 | 17.9 |
|  | 31–35 years | 54 | 19.4 |
|  | 36–40 years | 87 | 31.2 |
|  | 41–45 years | 45 | 16.1 |
|  | 46–50 years | 17 | 6.1 |
|  | >51 years | 19 | 6.8 |
| Education Level | Junior high school | 2 | 0.7 |
|  | Senior high school | 68 | 24.4 |
|  | Junior college | 71 | 25.4 |
|  | University | 94 | 33.7 |
|  | Masters | 37 | 13.3 |
|  | Doctorate/PhD | 7 | 2.5 |
| Years of Operation | ≤3 | 44 | 15.8 |
|  | 3~6 | 60 | 21.5 |
|  | 6~9 | 56 | 20.1 |
|  | 9~12 | 35 | 12.5 |
|  | 12~15 | 30 | 10.8 |
|  | ≥15 | 54 | 19.4 |
| Organizational Type | Other | 3 | 1.1 |
|  | Chain operation | 123 | 44.1 |
|  | Independent operation | 153 | 54.8 |

**Table 2.** *Cont.*

| Item | Class | Number (People) | Percent (%) |
|---|---|---|---|
| | Cashier | 2 | 0.7 |
| | First-line supervisor | 1 | 0.4 |
| Job Titles | Restaurateur | 113 | 40.5 |
| | Manager | 162 | 58.1 |
| | Chief | 1 | 0.4 |

The ratio of male to female respondent is similar, with 50.9% male and 49.1% female. Education level wise, university stood the most at 33.7%, followed second by college at 25.4%, then high school education level at 24.4%, and middle school education level was the least at 0.7%. In total, 31.2% of the respondents aged between 36 to 40 years old stood the most, followed by 19.4% of the respondents aged between 31 to 25 years old, and the least were respondents below 25 years old at 2.5%. Food- and beverage-related work experience between 3 to 6 years stood the most at 21.5%, followed by those with 6 to 9 years of work experience at 20.1%, and the least were those with 12 to 15 years of work experience at 10.8%. Respondents who were management level stood at 58.1%, and business operators stood at 40.5%.

The descriptive statistics for each question and dimension are summarized in Table 3.

**Table 3.** The descriptive statistics of constructs and items.

| Item | Mean | SD |
|---|---|---|
| Sales Growth | 3.928 | 0.772 |
| 1. Mobile payments are compatible with other payment options used in our company. | 3.996 | 0.923 |
| 2. Mobile payments are compatible with our company's work routines. | 3.878 | 1.035 |
| 3. Our company's products are applicable to be paid for with mobile payments. | 4.047 | 0.990 |
| 4. We wish that our customers use mobile payments. | 3.918 | 1.027 |
| 5. Companies that offer mobile payments are forerunners. | 3.878 | 1.049 |
| 6. Offering mobile payments enhances our company's image among customers. | 3.910 | 0.942 |
| 7. Offering mobile payments increases our appreciation by other companies in our business. | 3.871 | 0.912 |
| Cost saving | 3.790 | 0.768 |
| 8. Mobile payments decrease our company's costs. | 3.459 | 1.114 |
| 9. Paying with a mobile phone speeds up payments. | 4.036 | 0.944 |
| 10. Paying with a mobile phone is efficient. | 4.125 | 0.903 |
| 11. Mobile payments free resources for other purposes. | 3.577 | 1.172 |
| 12. Mobile payments make the processing of complaints easier. | 3.530 | 1.017 |
| 13. Mobile payments help the staff to concentrate on more important tasks. | 3.703 | 0.986 |
| 14. Mobile payments are useful. | 4.097 | 0.783 |
| Flexibility | 3.857 | 0.792 |
| 15. Mobile payments increase impulse purchases. | 3.728 | 1.034 |
| 16. Mobile payment benefits include the ability of customers to pay independently of time. | 3.910 | 0.875 |
| 17. Mobile payment benefits include the ability of customers to pay independently of place. | 3.932 | 0.929 |
| Accessibility | 3.946 | 0.805 |
| 18. It is easy for the personnel to learn to use the mobile payment system. | 3.935 | 0.895 |
| 19. It is easy for the personnel to process mobile payments. | 3.957 | 0.920 |
| 20. Mobile payments are easy to understand. | 4.007 | 0.898 |
| 21. It is easy to pay with a mobile phone. | 4.122 | 0.848 |
| 22. It is easy to instruct customers on how to use mobile payments. | 3.710 | 0.909 |

**Table 3.** *Cont.*

| Item | Mean | SD |
|---|---|---|
| Trust and safety | 3.82 | 0.887 |
| 23. Cooperation partners, such as financial institutions and telecom operators, are trustworthy. | 3.961 | 0.934 |
| 24. Mobile payments are secure. | 3.753 | 1.003 |
| 25. Mobile phones are reliable enough for payment transactions. | 3.756 | 0.970 |
| 26. Mobile networks are reliable enough for payment transactions. | 3.810 | 0.950 |
| Performance | 4.064 | 0.708 |
| 27. Compared with your industry as a whole, how would you rate your organization's performance in terms of public image and goodwill? | 4.151 | 0.772 |
| 28. Compared with your industry as a whole, how would you rate your organization's performance in terms of the growth rate of sales or revenues? | 4.011 | 0.798 |
| 29. Compared with your industry as a whole, how would you rate your organization's performance in terms of product or service quality? | 4.057 | 0.803 |
| 30. Compared with your industry as a whole, how would you rate your organization's performance in terms of employee productivity? | 4.036 | 0.790 |

For the construct of sales growth, the average score is 3.928, and the standard deviation is 0.772. In this dimension, for question 3, "Your company's product is suitable for mobile payment", the average score is 4.407, which is higher than the overall average value, indicating the restaurant industries suitable for the mobile payment environment. In the second dimension of "cost savings", the overall average score is 3.79, and the standard deviation is 0.768. In this dimension, the scores of the top three questions were question #10, "The use of the mobile phone for payment is effective"; question #14, "Mobile payment is useful"; and question #9, "The use of mobile payment speeds up the payment process". This shows that respondents agree that using mobile payment at restaurants is effective. In dimension three, flexibility, the overall average score is 3.857, and the standard deviation is 0.792. In this dimension, three questions are approaching four points. This shows that respondents intend to agree that NFC mobile payments are helpful for flexibility. In dimension four, accessibility, the overall average is 3.946, and the standard deviation is 0.805. In this dimension, in only question 21, "the use of mobile payment is easy", and question 20, "it is easy to understand mobile payment", the average value surpasses 4 points, and the overall value is high. This shows that respondents think that the use of mobile payment is easy, and it is easy to understand. In dimension five, trust and safety, the overall value is 3.82, and the standard deviation is 0.887. This shows that the selected restaurant operators intentionally agree that the current mobile payment operation environment is secure. For the dimension "ROP", the average value is 4.064, and the standard deviation is 0.708. In this dimension, all of the questions have an average value of four points. This shows that respondents agree that the restaurants at which they work have a greater performance value than others in the same industry.

### 3.2. Moderated Regression Results

In order to investigate the relationship between relative advantage, flexibility, accessibility, and trust and safety toward ROP, linear multiple regression was first used. Researchers suggested that the regression model is better than the structure equation model (SEM) in exploratory studies [83,84]. Meanwhile, Bryne [87] indicated that each of the SEM softwares differ in the way they treat missing data, and many methods are available to users to handle incomplete data. Meanwhile, different software produced different types of fit indices. However, regression analysis using the SPSS program is properly direct and easier to use [88,89]. The moderated regression model was further used to examine the moderating effect of restaurant size between independent variables and ROP. The regression model explained 52 percent of the variation in the dependent variable ROP, as indicated by the adjusted-$R^2$ = 0.52 in Table 4. The variance inflation factor (VIF) range was

between 3.313 and 4.639. As the VIF value for the various dimensions was smaller than 10 [85], it was known that there were no serious collinearity problems between various dimensions. Five factors had a significant effect on ROP. These included sales growth ($\beta$ = 0.453; $p$ < 0.000); cost saving ($\beta$ = −0.236; $p$ = 0.009); flexibility ($\beta$ = 0.117; $p$ = 0.098); accessibility ($\beta$ = 0.184; $p$ = 0.035); and trust and safety ($\beta$ = 0.286; $p$ < 0.000).

**Table 4.** The regression results (N = 279).

| Independent Variables | $\beta$ | *p*-Value | VIF |
|---|---|---|---|
| Sales growth | 0.453 *** | 0.000 | 3.313 |
| Cost saving | −0.236 *** | 0.009 | 4.639 |
| Flexibility | 0.117 * | 0.098 | 2.869 |
| Accessibility | 0.184 ** | 0.035 | 4.378 |
| Trust and safety | 0.286 *** | 0.000 | 2.672 |
| $R^2$ = 0.529, Adjusted $R^2$ = 0.520, and F-value = 61.227 | | | |

Dependent variable: Restaurant operating performance; * $p$ < 0.1, ** $p$ < 0.05, and *** $p$ < 0.01.

The standardized beta values suggest that sales growth has the greatest impact on ROP. Trust and safety, accessibility, and flexibility were also determined to be a significantly positive impact on ROP. However, it should be noted that cost savings were negatively correlated, indicating that excessively reducing costs may have a detrimental impact on ROP. The food and beverage industry is labor-intensive, and excessive cost-cutting measures might compromise service quality, leading to a decline in customer satisfaction and a reluctance to revisit, ultimately affecting performance negatively. Therefore, cost management should be carefully balanced to maintain service quality and ensure customer satisfaction.

### 3.3. Moderation Effect

Moderated regression was conducted using interaction terms between independent variables and moderating variables. Model 1 is the main-effects model in Table 5. The size of the restaurant was added as a moderator; however, after the multi-collinearity test (see Table 5, Model 2), the VIF value of the interaction term of sales growth by restaurant size and the interaction term of cost saving by restaurant size were 36.649 and 48.269, indicating high collinearity between these two interaction terms. Therefore, these two interaction terms for sales × restaurant size and cost saving × restaurant size were deleted.

**Table 5.** The moderated regression (N = 279).

| Variable Name | Dependent Variable: ROP | | | |
|---|---|---|---|---|
| | Model 1 | | Model 2 | |
| | $\beta$ | VIF | $\beta$ | VIF |
| Sales growth | 0.453 *** | 3.313 | 0.499 *** | 3.613 |
| Cost saving | −0.236 *** | 4.639 | −0.247 *** | 5.938 |
| Flexibility | 0.117 * | 2.869 | 0.104 | 2.915 |
| Accessibility | 0.184 ** | 4.378 | 0.161 * | 4.691 |
| Trust and safety | 0.286 *** | 2.672 | 0.298 *** | 2.932 |
| Interactions | | | | |
| Sales growth × restaurant size | | | 0.298 | 36.649 |
| Cost saving × restaurant size | | | −0.161 | 48.269 |
| Flexibility × restaurant size | | | 0.042 | 1.498 |
| Accessibility × restaurant size | | | 0.099 | 9.290 |
| Trust and safety × restaurant size | | | −0.308 ** | 10.806 |
| $R^2$ | 0.529 | | 0.543 | |
| Adj $R^2$ | 0.52 | | 0.526 | |
| F | 61.227 | | 39.595 | |
| df | (5, 273) | | (8, 270) | |

Dependent variable: Restaurant operating performance.* $p$ < 0.1, ** $p$ < 0.05, and *** $p$ < 0.01.

Model 3 in Table 6 was revised to three interaction terms because of the elimination of collinearity problems. The adjusted $R^2$ was 0.526 of model 3 in Table 6, indicating that this regression model explained 52.6 percent of the variation in the dependent variable ROP. After accounting for differences in restaurant size as a moderator, the accessibility and trust and safety in ROP vary on the basis of restaurant size. Two interactions were statistically significant. The interaction term between accessibility and restaurant size had a positive impact (β = 0.108) on ROP, suggesting that larger restaurants benefitted from the easier utilization of NFC mobile payment, leading to improved ROP. This finding supported the idea that larger establishments require more efficient equipment to streamline complex processes. However, the interaction term between trust and safety and restaurant size showed a negative impact (β = −0.169) on ROP. This suggested that an excessive number of personnel handling the mobile payment system may create checkout risks, leading to consumer distrust and ultimately resulting in poorer ROP.

**Table 6.** The moderated regression results after deletion of collinear variables.

| Variable Name | Dependent Variable: ROP | | | |
| --- | --- | --- | --- | --- |
| | Model 1 | | Model 3 | |
| | β | VIF | β | VIF |
| Sales growth | 0.453 *** | 3.313 | 0.478 *** | 3.372 |
| Cost saving | −0.236 *** | 4.639 | −0.236 *** | 4.667 |
| Flexibility | 0.117 * | 2.869 | 0.103 ** | 2.914 |
| Accessibility | 0.184 ** | 4.378 | 0.163 | 4.422 |
| Trust and safety | 0.286 *** | 2.672 | 0.307 * | 2.717 |
| Interactions | | | | |
| Flexibility × restaurant size | | | 0.020 | 1.317 |
| Accessibility × restaurant size | | | 0.108 * | 2.448 |
| Trust and safety × restaurant size | | | −0.169 ** | 2.592 |
| $R^2$ | 0.529 | | 0.54 | |
| Adj $R^2$ | 0.52 | | 0.526 | |
| F | 61.227 | | 39.595 | |
| df | (5, 273) | | (8, 270) | |

Note: ROP: Restaurant operating performance; * $p < 0.1$, ** $p < 0.05$, and *** $p < 0.01$.

## 4. Discussion

As shown in Model 1 in Table 7, in addition to cost saving, shown with a negative correlation, the remaining four dimensions are positively correlated. The hypothetical empirical result of this research is shown in Table 7. In terms of H1 and H2, the supported hypotheses are consistent with the study of literature [59]. Sheikh et al. [59] noted that the relative advantage environment positively influences ROP. In addition, this cost-saving result is consistent with the findings of Moghavvemi et al. [12] and Mallat and Tuunainen [77]. The high cost of investing in mobile payment infrastructure acts as a barrier for merchants considering the adoption of mobile payments. Mallat and Tuunainen [77] observed that while businesses acknowledge the advantages of NFC mobile payments, the initial setup costs are substantial, and uncertainties about the timing of significant cost reductions could potentially account for the adverse correlation observed in this study within the cost-saving dimension. Additionally, the restaurant industry relies heavily on manpower, and excessive cost-cutting measures could compromise service quality, negatively impacting operational performance. Fang [78] further indicated that food and labor cost efficiency are the two most important factors for the operation of restaurants.

In terms of H3, this finding was consistent with the work of previous literature. Flexibility is also recognized as one of the most important factors for the success of the mobile commerce environment [63]. In terms of H4, accessibility had a positive impact

on ROP. This result was consistent with the work of Dahlberg et al. [66]. Their study also noted that accessibility is the most crucial factor in the mobile payment context.

**Table 7.** The empirical results of the research hypothesis.

| Variable Name | Model 1 | | Model 3 | |
|---|---|---|---|---|
| | β | Decision | β | Decision |
| Sales growth | 0.453 *** | H1 Supported | 0.478 *** | |
| Cost saving | −0.236 *** | H2 Supported | −0.236 *** | |
| Flexibility | 0.117 * | H3 Supported | 0.103 ** | |
| Accessibility | 0.184 ** | H4 Supported | 0.163 | |
| Trust and safety | 0.286 *** | H5 Supported | 0.307 * | |
| Interactions | | | | |
| Flexibility × restaurant size | | | 0.020 | H6c Rejected |
| Accessibility × restaurant size | | | 0.108 * | H6d Supported |
| Trust and safety × restaurant size | | | −0.169 ** | H6e Supported |

\* $p < 0.1$, \*\* $p < 0.05$, and \*\*\* $p < 0.01$.

In terms of H5, this study confirmed that the trust and safety construct positively impacts ROP. This empirical finding was in agreement with the outcomes observed in the research conducted by Moghavvemi et al. [12]. Moghavvemi et al. [12] conducted in-depth interviews with merchants across various retail categories, aiming to delve into the motivations and obstacles concerning the adoption of mobile payment systems in Malaysia. The improved payment security feature serves as a motivating factor for merchants when considering mobile payments.

In terms of moderating effect, as shown in model 3 in Table 7, the hypothesis of the moderation effect of restaurant size between accessibility and ROP and trust and safety and ROP was supported. This result was partially consistent with the works from the literature. Melitz and Ottaviano [72] and Rumelt [73] suggested that larger enterprises have certain advantages over smaller companies in terms of market power, management processes, and financial resources. However, concerning safety and trust aspects, larger companies might face security risks in the mobile payment processes.

## 5. Conclusions, Implications, and Limitations

### 5.1. Conclusions

This study integrated IDT with TAM to examine the impact of NFC mobile payments on ROP. This research invited eight experts from industry and academia to test the validity of the questionnaire. Reliability testing of the questionnaires was conducted through the distribution of 52 hard copies of the valid questionnaire. This paper used convenience sampling to distribute questionnaires to restaurant owners and managers instead of customers in order to investigate the impact of NFC mobile payment on ROP. A total of 279 valid questionnaires were collected. Multiple regression analysis was used to explore the relationship between independent and dependent variables. The research results showed that respondents believed the five dimensions (sales growth, cost saving, flexibility, accessibility, and trust and safety) had a significant influence on ROP, but the cost-saving dimension had a significantly negative impact on ROP. After considering the restaurant size as a moderator for analysis, there were two constructs, "accessibility" and "trust and safety", which had significantly positive impacts on ROP.

### 5.2. Theoretical Implications

Early research on mobile payments mostly focus on the ubiquity and personal traits of the system and services [90]. This research was based on the integration of IDT and TAM theories, added to two recognized key variables (flexibility and trust and safety). Through empirical research of restaurateurs and catering managers on the impacts of mobile payment on operations performance, this research proves that the features of NFC

mobile payment, including relative advantages, flexibility, accessibility, and trust and safety, can effectively increase the performance of restaurant operations. These empirical results provide the theoretical implications of the integration of IDT and TAM, and the other two constructs (flexibility and trust and safety) could explain the impact of NFC mobile payments on operational performance in the restaurant industry. This result was consistent with the result of research conducted by Mallat and Tuunainen [77] through a questionnaire survey and semi-structured interview. For the NFC payment environment in Taiwan, the empirical results of this research can provide relevant researchers to further develop and improve the theoretical model of NFC mobile payment. This can also provide a more specific and effective payment system for restaurant managers.

Meanwhile, this research examined the moderation effect of the size of the restaurant to mobile payment on operational performance. The analysis showed that only the interaction term of accessibility and restaurant size had a significantly positive moderation effect, and another interaction term of trust and safety and restaurant size had a significantly negative moderation effect. This result was different from the research results of Melitz and Ottaviano [72] and Rumelt [73]. From the management perspective, the possible reason for inference was that the bigger companies have more diversified partners, such as financial institutions and telecom operators, leading to involved complex processes to mitigate the payment process risk through more investment. Hence, as the size of the company is larger, more investment yields less operational performance.

*5.3. Practical Implications*

Amidst the government's active promotion, mobile payment applications have rapidly emerged in various fields in Taiwan over the past year. However, the data from this research indicate that the average rating for each dimension has not reached the 4-point level. This suggests that although mobile payment is a future trend, business operators have not yet reaped the expected benefits. They have yet to fully enjoy the advantages of mobile payments. Based on the analysis results, this research offers managerial practice recommendations to serve as decision-making references for mobile payment service providers.

The findings highlight trust and safety as the second most important construct for operational performance. Previous literature has emphasized the significance of transaction security in NFC mobile payment, with consumers expressing concerns about potential risks [67,71]. To enhance consumer confidence, mobile payment service providers can explore additional verification mechanisms like biometrics (iris, fingerprint) to strengthen transaction security and increase user willingness to adopt the technology.

Furthermore, ensuring system stability and data safety is crucial to prevent the leakage of personal information or system instability. Any breach in these aspects can lead to user distrust, emphasizing the importance for businesses to take extra precautions and fulfill their responsibilities in implementing robust security measures.

It is known from past literature and information that NFC mobile payment is one of the trends in future payment method development [91,92]. But, cash and credit card are still the major forms of payment. Currently, the use of NFC mobile payments is not as simple as making payments using cash or credit card. For a new type of payment method, it will certainly undergo comparison by consumers or frontline operating employees with past usage habits. If a better usage experience is not achieved, consumers will not be willing to use the new payment method, and frontline-operating staff will also refuse to use it. This will be a huge obstacle in the promotion of the method. Therefore, mobile payment service providers should start by simplifying the operating procedures, reducing operating time for the users, and reducing barriers to learning or use [93]. In any innovation, it must start with a good consumer payment experience, thus raising payment convenience and reducing the pain for users. Operators must provide a high-quality mobile payment system so that consumers can use it anytime and anywhere. Mobile payment can then be a part of regular activities in daily living, and it will allow profits for operators at the same time.

Promoting mobile payments is going to be a lot easier only when either side can profit from this.

Finally, this research suggests that one can make use of the backend big data of the mobile payment to combine with the advertising activities promoted by the food and beverage businesses on the APP. This will lower the advertisement costs for the businesses, and consumers can receive firsthand discount ad information and boost consumption. On the other hand, when consumers continue to use mobile payment, the financial industry can also earn a profit on the payment fees, and the three parties benefit.

This paper also provides the policymakers with some suggestions. (1) Regarding the reason affecting the construction of mobile payment, there is the cost factor. There are newly added taxes when business operators generate invoices, fees charged by the third-party payment industry, high construction costs, and more in Taiwan. The lack of an appropriate fee-charging model for adopting mobile payment is one of the primary problems that needs to be resolved first. (2) On the technical side, the problem is the failure to integrate the end system for mobile payment. Methods to integrate QR codes, sensors, and other forms should be actively sought after. Currently, mobile payment service providers are developing their own special tools for designated stores. Consumers often see three or four types of POS machines placed on the cashier counter upon entering these stores. If the consumer experiences poor-performing sensor moments during mobile payment several times, they would end up using a payment method that they are more accustomed to.

Taiwan's mobile payment penetration rate is behind China, India, and even continents like Africa. Although the government has developed mobile payments and listed this as one of the primary policy objectives, it is still far from the 2025 goal of attaining a 90% penetration rate as set by the Financial Supervisory Commission Taiwan [94]. The goal would not be met unless the operators' demand for those problems is addressed and resolved.

*5.4. Research Limitations and Future Research*

The study examines the effects of five constructs and one moderator on restaurant performance from the perspectives of restaurant owners and management-level staff. Exploring similar impacts in other hospitality industries, such as hotels and retailers, could be worthwhile. However, this paper acknowledges limitations, as it employs TAM and IDT to investigate the applicability of mobile payments in restaurant operations. Nevertheless, certain variables, such as eat-in or takeaway options and meal prices, may have a direct or indirect influence on the outcomes. Future research could incorporate additional variables to provide a more comprehensive understanding of real-world scenarios.

This paper focused on gathering opinions from restaurateurs, and due to the difficulty in requesting them to fill in questionnaires, convenience sampling was employed. For future research, quota sampling could be considered to enhance the generalizability of the findings.

For future research, expanding the sample size by including perspectives from both merchants and customers is recommended. This will enable the investigation of other factors influencing merchants' decisions to adopt mobile payment systems. Additionally, conducting a questionnaire survey with consumers can help identify any gaps between merchants and customers, leading to the development of improvement plans and potentially increasing the adoption rate of mobile payments in Taiwan.

Regarding the analytical approach, Gefen et al. [95] pointed out that SEM has better explanatory power than regression analysis, based on the theoretical foundation of this research. However, in exploratory studies, researchers have suggested that regression analysis using the SPSS program may be more straightforward and user-friendly [76,77]. For enhanced explanatory power of variables, future studies can consider using SEM with different software as an analytical tool.

The respondents for both the independent and dependent variables of the two dimensions are the same, which may introduce systematic bias due to measurement methods,

potentially affecting the conclusions. To mitigate this, future research is advised to match consumers and industry operators as respondents to avoid potential common method variance (CMV) issues.

**Author Contributions:** Conceptualization, W.-W.Y.; Methodology, C.-Y.F.; Investigation, W.-W.Y.; Data curation, W.-W.Y.; Writing – original draft, C.-Y.F.; Writing—review & editing, C.-Y.F.; Supervision, C.-Y.F.; Project administration, C.-Y.F.; Funding acquisition, C.-Y.F. All authors have read and agreed to the published version of the manuscript.

**Funding:** This work was supported by Taiwan's National Science and Technology Council (MOST 110-2622-H-003-010), (NSTC 112-2410-H-003-129) and the National Taiwan Normal University, Taiwan.

**Data Availability Statement:** The datasets used and/or analyzed during the current study are available from the corresponding author.

**Conflicts of Interest:** The authors declare no conflict of interest.

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
