# Peer review of "The Role of Near-Field Communication Mobile Payments in Sustainable Restaurant Operations: A Restaurateur’s Perspective"

_sustainability, doi:10.3390/su151612471_

Round 1
Reviewer 1 Report
Dear Authors, thank you for letting me review your article.
My comments:
- to introduction: in some parts, the introduction is somewhat scattered and could be more focused on the main objectives of the research. It is also worth noting the linguistic accuracy, especially in some sentences that may require slight rephrasing.
The research method described in the text appears to be appropriate and suitable for the research objectives. The use of a questionnaire and Likert scales as measurement tools is commonly employed in scientific studies.
It is important that the study was preceded by a pretest, which involved testing the questionnaire to assess its accuracy and comprehensibility. Engaging experts from various fields to validate the questionnaire is a good research practice.
Furthermore, conducting a reliability test using Cronbach's alpha coefficient is significant. The Cronbach's alpha values for the individual dimensions of the research fall within an acceptable range, indicating their consistency and reliability.
no futher suggestions
Author Response
Response to Reviewers’ Comments
Manuscript ID: Sustainability-2516036
Title: Embracing Sustainability: Assessing the Influence of Near-Field Communication Mobile Payment Systems on Restaurant Operating Performance
The authors would like to express our great appreciation to the referee for the helpful and constructive comments on our work and the manuscript of the paper.
This paper has been carefully revised in accordance with your helpful suggestions. The changes are listed in red in the following as well as shown in the revised paper in the attachment as followed.
Reviewer 1
- to introduction: in some parts, the introduction is somewhat scattered and could be more focused on the main objectives of the research. It is also worth noting the linguistic accuracy, especially in some sentences that may require slight rephrasing.
Author Response:
Thank you for the reviewer's comment. We have sought the assistance of a native speaker to edit our paper, and we have highlighted the revised parts in red.
- The research method described in the text appears to be appropriate and suitable for the research objectives. The use of a questionnaire and Likert scales as measurement tools is commonly employed in scientific studies. It is important that the study was preceded by a pretest, which involved testing the questionnaire to assess its accuracy and comprehensibility. Engaging experts from various fields to validate the questionnaire is a good research practice. Furthermore, conducting a reliability test using Cronbach's alpha coefficient is significant. The Cronbach's alpha values for the individual dimensions of the research fall within an acceptable range, indicating their consistency and reliability.
Author Response:
We appreciate the reviewer's positive feedback.

Reviewer 2 Report
The main question addressed by the research is the impact of near-field-communication (NFC) mobile payment systems on restaurant operating performance (ROP). The topic is relevant in the field as it addresses the potential benefits of adopting NFC mobile payment systems in restaurants, which can improve sales growth, cost savings, flexibility, accessibility, and trust & safety. It also identifies the negative impact of cost-saving dimension on ROP. The study adds to the subject area by examining the drivers and barriers for merchants using NFC mobile payment environments and investigating the impact of NFC mobile payment on ROP.
The study adds to the subject area by integrating the Information System Success Model (ISSM) with the Technology Acceptance Model (TAM) and Innovation Diffusion Theory (IDT) to examine the impact of NFC mobile payments on ROP. It also identifies the specific dimensions that have a significant influence on ROP and the moderating effect of restaurant size on the relationship between independent and dependent variables.
The authors could consider using a larger sample size and a more diverse group of participants to increase the generalizability of the findings. They could also consider using a longitudinal design to examine the long-term impact of NFC mobile payment systems on ROP.
The conclusions are consistent with the evidence and arguments presented and address the main question posed. The study provides valuable insights into the benefits of adopting NFC mobile payment systems in restaurants and identifies the specific dimensions that have a significant influence on ROP.
The references are appropriate and include relevant studies in the field.
However, here are some points that need justification and some suggestions for the paper:
· Clarification on the sampling method used and how representative the sample is of the population of interest.
· Further explanation of the constructs used in the study and how they were measured.
· Justification for the use of convenience sampling and how it may affect the generalizability of the findings.
· More detailed information on the experts who tested the validity of the questionnaire and how their feedback was incorporated into the final version.
· Further discussion on the negative impact of the cost-saving dimension on ROP and how it can be addressed.
· More detailed information on the moderating effect of restaurant size on the relationship between independent and dependent variables.
· Further discussion on the limitations of the study and how they can be addressed in future research.
Here are some suggestions in the introduction and literature review sections that may require moderate editing:
- Sentence structure: Some sentences in the introduction section are quite long and complex, which can make them difficult to read and understand. Consider breaking them up into shorter, more concise sentences to improve clarity and readability.
- Word choice: There are a few instances where the word choice could be improved to better convey the intended meaning.
- Grammar: There are a few instances where the grammar could be improved to ensure proper subject-verb agreement and sentence structure.
- Clarity: Some sentences could be rephrased to improve clarity and avoid ambiguity.
Reviewer 3 Report
The article ‘Embracing Sustainability: Assessing the Influence of Near-Field Communication Mobile Payment Systems on Restaurant Operating Performance’’ is interested article, however, it require major changes. Moreover the technical, novel side of the paper is very week. Following are the comments for the authors to improve the article:
Ø The article title is a general statement, authors are suggested to re-write the title with the research impact and novelty perspective.
Ø Also please avoid general information in the title.
Ø A lot of research is going on Near-Field Communication. How authors claim the new aspect for the need of this publication?
Ø Problem statement should be mentioned at the start of the abstract, it is missing. Why the study is important? What is the novelty aspect?
Ø What is the base of selection of this study? Author should define some specific criteria.
Ø Section of literature review should be merged in Introduction section instead of separate section.
Ø The keywords should be specific.
Ø Line 181: Cumulative references should be avoided i.e [47, 51, 59, 60]. There should be only 1-2 references. Instead of merging references, authors should add more up to date and state of the art literature, if possible.
Ø There should be some statistical figured values in the abstract which can quantify the research / optimization and it can make readership of the journal easy.
Ø There should be some proper synchronization of the sentences in meaningful way.
Ø The abstract should also include the solution of the problem based on the problem statement with some particular application/s.
Ø Also, the literature review section is very long. It should be reduced.
Ø Table 1 Validity Test: the captions should be descriptive and specific.
Ø There are few old references, authors are encouraged to add latest literature.
Ø There are same type of results presented. Paper need data analysis on the results from different perspectives.
Ø Conclusion and implications section is very long. It should be compact and straightforward.
minor
Round 2
Reviewer 3 Report
accept
this is ok
Author Response
Thank you for your recognition.